# Review on Photoacoustic Monitoring after Drug Delivery: From Label-Free Biomarkers to Pharmacokinetics Agents

**DOI:** 10.3390/pharmaceutics16101240

**Published:** 2024-09-24

**Authors:** Jiwoong Kim, Seongwook Choi, Chulhong Kim, Jeesu Kim, Byullee Park

**Affiliations:** 1Departments of Electrical Engineering, Convergence IT Engineering, Medical Science and Engineering, Mechanical Engineering, and Medical Device Innovation Center, Pohang University of Science and Technology (POSTECH), Cheongam-ro 77, Nam-gu, Pohang 37673, Republic of Korea; dndtm9823@postech.ac.kr (J.K.); swchoi715@postech.ac.kr (S.C.); chulhong@postech.edu (C.K.); 2Departments of Cogno-Mechatronics Engineering and Optics & Mechatronics Engineering, Pusan National University, Busan 46241, Republic of Korea; 3Department of Biophysics, Institute of Quantum Biophysics, Sungkyunkwan University, Suwon 16419, Republic of Korea

**Keywords:** photoacoustics, drug delivery, treatment

## Abstract

Photoacoustic imaging (PAI) is an emerging noninvasive and label-free method for capturing the vasculature, hemodynamics, and physiological responses following drug delivery. PAI combines the advantages of optical and acoustic imaging to provide high-resolution images with multiparametric information. In recent decades, PAI’s abilities have been used to determine reactivity after the administration of various drugs. This study investigates photoacoustic imaging as a label-free method of monitoring drug delivery responses by observing changes in the vascular system and oxygen saturation levels across various biological tissues. In addition, we discuss photoacoustic studies that monitor the biodistribution and pharmacokinetics of exogenous contrast agents, offering contrast-enhanced imaging of diseased regions. Finally, we demonstrate the crucial role of photoacoustic imaging in understanding drug delivery mechanisms and treatment processes.

## 1. Introduction

In recent decades, theragnostic techniques that combine diagnosis and therapy have gained significant attention [1,2,3,4,5]. Various biomedical imaging modalities have been used to evaluate the functional capabilities of diagnostic and therapeutic agents [6,7,8,9,10]. Conventional imaging techniques, including computed tomography (CT) [11,12,13], magnetic resonance imaging (MRI) [14,15,16], and positron emission tomography (PET) [17,18] are useful for the three-dimensional visualization of drug reactions throughout the whole body. However, their high cost and large size may limit their efficiency for the routine monitoring of drug delivery.

Optical imaging techniques such as fluorescence imaging (FLI) and two-photon microscopy (TPM), have also been used to monitor drug delivery [19,20,21,22,23]. Compared to previous imaging techniques, optical imaging is less expensive, easier to implement, has faster imaging speed, and does not use ionizing radiation [24,25,26]. However, optical imaging has limitations, including restricted imaging depth, low spatial resolution, and limited anatomical information, making it unsuitable for deeper tissues and more detailed structural studies [27].

Photoacoustic imaging (PAI) is an advanced imaging technique based on the photoacoustic (PA) effect that involves energy transformation via light absorption and subsequent heat release [28,29,30,31]. PAI, which combines the advantages of optical and ultrasound imaging (USI), provides molecular functional information with fine spatial resolution in relatively deep tissue [32,33,34,35,36,37,38]. PAI offers high-resolution imaging with ranges typically in the under tens of micrometers to hundreds of micrometers (depending on the PAI system), and penetration depths over the optical diffusion limit (~1 mm), reaching up to several centimeters [39,40,41,42]. Additionally, PAI is a powerful tool for the label-free assessment of hemodynamics and physiologies, including hemoglobin oxygen saturation (sO_2_; Abbreviations and acronyms frequently used in this paper are shown in Table 1) levels, which indicate the metabolic response of biological tissues. This is because PAI uses hemoglobin as an endogenous contrast agent, allowing for the simultaneous acquisition of both morphological vascular images and corresponding sO_2_ information [43,44], which plays a crucial role in the application of PAI compared to other modalities. Using the distinct optical absorption spectra of oxy-hemoglobin (HbO) and deoxy-hemoglobin (HbR), multispectral PAI can provide sO_2_ distribution without injecting exogenous agents [45,46,47,48,49,50]. The feasibility of label-free sO_2_ analysis using PAI has been demonstrated in numerous biomedical studies [51,52,53,54]. This method has been particularly investigated for human clinical applications [54,55,56,57,58,59,60,61,62], where injecting agents is challenging. Using this technique, PAI can be used to assess variations in sO_2_ levels in tissues following drug delivery. Furthermore, various agents have been developed and studied [63,64,65,66,67,68,69]. When therapeutic functions are added to these agents, contrast-enhanced PAI can monitor drug delivery and treatment responses [70,71,72,73,74,75,76,77,78,79].

In this study, we review the comprehensive achievements of PAI by investigating the biological responses following drug delivery (Figure 1). First, we discuss the principles of PAI, including label-free imaging and multispectral responses of biological tissues. Subsequently, we review studies that evaluated the morphological, hemodynamic, and physiological reactions following drug delivery based on label-free PAI. This section provides a comprehensive overview of the analysis of vascular system responses to drug delivery across various tissues in both preclinical and clinical stages. PAI-based monitoring following drug delivery aims to provide future directions for diagnostic and therapeutic applications, particularly for assessing the agent delivery efficiency and treatment efficacy of phototherapies, thereby evaluating the potential clinical translation of these therapeutic agents. Finally, we highlight the representative state-of-the-art results for monitoring biodistribution and pharmacokinetics by directly detecting administered contrast agents. This suggests that by adding functions such as disease targeting to the exogenous contrast, PAI can be used to monitor pharmacokinetics, diagnosis, and therapeutic effects.

## 2. Background and Principle

### 2.1. Principle of Photoacoustic Imaging

PAI uses the PA effect, which converts absorbed light into acoustic waves. When a short laser pulse illuminates biological tissues, specific chromophores (light-absorbing molecules) absorb the optical energy. The absorption of light causes rapid thermal expansion in chromophores, which increases localized pressure. The initial pressure ($P_{o}$) can be defined as follows:(1)$$P_{o}=\Gamma \cdot \sigma \cdot \mu_{a}\cdot F,$$
where Γ denotes the Gruneisen parameter (which relates temperature change to pressure change), σ denotes the heat conversion efficiency, $\mu_{a}$ denotes the optical absorption coefficient, and F denotes the local fluence (optical energy per unit area). These factors determine how efficiently absorbed light energy is converted into pressure. Therefore, variations in light intensity, pigment concentration, or these parameters directly affect the PA signal’s magnitude [87], demonstrating the proportional relationship between $P_{0}$ and the PA response. Rapid thermal expansion generates ultrasound (US) waves, which are PA waves that propagate through tissues.

A typical PAI system comprises an optical source, detectors, and a data acquisition (DAQ) system [88]. The optical source, typically a pulsed laser, provides excitations, and its beam shape can be optimized using lenses or diffusers to ensure effective light delivery to the target tissue. The generated PA signals are then detected by a range of detectors selected based on specific requirements, such as a conventional piezoelectric ultrasound transducer, a piezoelectric micromachined ultrasonic transducer, or optical sensors [32]. The DAQ system processes these electrical signals, converting them into usable images [89]. This involves digitizing, filtering, demodulating, and amplifying the signals, followed by image reconstruction using various algorithms such as delay-and-sum, filtered back-projection, or time-reversal. PAI configurations may also incorporate additional components like preamplifiers and multiplexer boards, or further signal and image processing tools, tailored to the particular application and its requirements [90].

### 2.2. Contrast of Photoacoustic Imaging

PAI can be performed using two main contrast mechanisms: label-free and labeled, with each providing distinct contrast mechanisms based on the presence of endogenous or exogenous chromophores [64,91,92]. In label-free PAI, endogenous contrast agents naturally present in tissues such as hemoglobin, melanin, and lipids provide the required optical absorption. These endogenous chromophores absorb specific wavelengths of light and produce PA signals without requiring external contrast agents. This technique is effective for imaging blood vessels, melanin-rich structures, and other tissue components that have inherent optical contrast [93]. For example, PAI can be used to assess wound healing by observing angiogenesis around a disease site and evaluating malignancy by measuring the size of the melanoma. Although label-free PAI has enormous potential, its limited endogenous contrast severely limits its use in preclinical and clinical settings. In contrast, labeled PAI involves the use of exogenous contrast agents to enhance optical absorption. These agents can include dyes, metallic nanoparticles, polymeric nanoparticles, and other materials with strong optical absorption at specific wavelengths [70]. When these labeled agents are administered to the tissue, they accumulate in specific regions, boosting the PA signal. Targeted imaging enables the high-contrast visualization of specific biological structures or molecular targets. Exogenous contrast-enhanced PAI has numerous applications, including visualizing and monitoring tumors, encephalitis, lymph nodes, the gastrointestinal tract, atherosclerotic plaques, protein-specific glycosylation, and the liver. Recent advances in PA contrast agents have increased their utility by combining imaging capabilities with therapeutic drugs, allowing for treatments such as photothermal therapy (PTT) and photodynamic therapy (PDT).

### 2.3. Multispectral Photoacoustic Imaging

Multispectral PAI uses the spectral characteristics of different chromophores to separate their contributions to the PA signal, providing detailed information regarding the molecular composition of the tissue [94]. In multispectral PAI, tissue is sequentially illuminated with multiple wavelengths of light. Different chromophores exhibit unique absorption spectra, indicating that they absorb varying amounts of light at different wavelengths. Spectral unmixing is a computational technique for segregating signals from different chromophores based on their distinct absorption characteristics by solving a system of linear equations. The mathematical representation of spectral unmixing is as follows: When the PA signals $s_{j}\left(\lambda_{i}\right)$ are measured at the *j*-th chromophore at wavelength $\lambda_{i}$, $S(\lambda_{i})$ are measured at N different chromophores at wavelength $\lambda_{i}$, and the signals can be expressed as follows [95]:(2)$$s_{j}\left(\lambda_{i}\right)\;\propto \;\mu_{a}^{j}\left(\lambda_{i}\right)\cdot C_{j},$$
(3)$$S\left(\lambda_{i}\right)=\sum_{j=1}^{N}s_{j}(\lambda_{i}),$$
where $\mu_{a}^{j}(\lambda_{i})$ denotes the absorption coefficient of the j-th chromophore at wavelength $\lambda_{i}$, and $C_{j}$ denotes the concentration of the j-th chromophore. By solving this equation, the concentrations $C_{j}$ of different chromophores can be determined, enabling detailed quantitative imaging of the molecular composition of the tissue.

Using spectral unmixing, sO_2_ in the blood can be calculated using the different absorption spectra of oxy-hemoglobin and deoxy-hemoglobin. The steps to calculate sO_2_ are as follows: First, PA signals from blood vessels are acquired at multiple wavelengths where the absorption spectra of oxy-hemoglobin and deoxy-hemoglobin differ significantly. Second, spectral unmixing is applied to determine the concentrations of oxy-hemoglobin ($C_{HbO}$) and deoxy-hemoglobin ($C_{HbR}$) at each pixel or voxel in the blood vessel image. Third, the total hemoglobin (HbT) concentration is calculated as $C_{HbT}$ = $C_{HbO}$ + $C_{HbR}$. Finally, the sO_2_ levels are calculated as follows:(4)$${sO}_{2}\left(\%\right)=\frac{C_{HbO}}{C_{HbT}}\times 100,$$

By incorporating this spectral unmixing method into PAI, it is possible to generate high-resolution functional images depicting tissue oxygenation, which is crucial for various biomedical applications such as tumor characterization and the monitoring of therapeutic responses.

In contrast, labeled PAI involves the use of exogenous contrast agents, such as dyes or nanoparticles, to enhance optical absorption. In this case, the administered agents are directly detected by PAI, and these agents are chosen for their strong absorption at specific wavelength ranges, which must be clearly distinguishable from the absorption characteristics of endogenous contrast agents such as oxy-hemoglobin and deoxy-hemoglobin. Additionally, it is necessary to consider other factors such as the concentration of the agent and its chemical modifications within the body when applying these PAI techniques. Similarly, as in label-free PAI, the PA signal from exogenous agents also can be separated from the background blood signal using spectral unmixing. This method allows for the extraction of the signals of the administered agents, enabling more accurate quantitative imaging. For effective PAI, photothermally active samples should possess key characteristics such as a high absorption coefficient, an optimized Gruneisen parameter, and distinct spectral characteristics. Additionally, photostability, biocompatibility, and appropriate concentration and distribution are essential to ensure strong signal generation and reliable imaging.

### 2.4. Photoacoustic Imaging Systems

PAI systems exhibit varying performance depending on their purpose and system configurations. These systems are broadly classified into photoacoustic microscopy (PAM) and photoacoustic computed tomography (PACT), depending on whether the detector uses a single-element transducer or multiple elements. First, PAM provides high-resolution, high-contrast images ranging from sub-micrometers to tens of micrometers, making it suitable for observing fine details [96]. PAM is further categorized into optical-resolution photoacoustic microscopy (OR-PAM) and acoustic-resolution photoacoustic microscopy (AR-PAM) based on the arrangement of acoustic detection and optical illumination. The differences in imaging performance such as lateral resolution and depth are largely influenced by this classification [97,98]. OR-PAM achieves high lateral resolution, often in the sub-micrometer to micrometer range by tightly focusing the optical beam [99]. However, OR-PAM’s imaging depth is limited to approximately 1 mm due to light scattering, making it ideal for capturing fine details in superficial layers, where high-resolution imaging is required. In contrast, AR-PAM’s lateral resolution is determined by the acoustic focus and typically ranges in tens of micrometers, allowing deeper penetration depths due to the lower scattering of acoustic waves in biological tissues. This makes AR-PAM suitable for imaging deeper structures in biological tissues, up to several millimeters [100,101]. On the other hand, PACT offers a greater imaging depth, speed, and larger field of view, along with resolution in the hundreds of micrometers. This makes PACT suitable for various preclinical and clinical applications that require large-field imaging [80,102,103]. Table 2 shows examples of the current performance of conventional PAI systems based on their classification.

## 3. Label-Free Photoacoustic Monitoring of Responses to Drug Delivery

The label-free imaging capability of PAI facilitates the observation of morphology, hemodynamics, and physiology following drug delivery. PAI has been extensively used to observe blood vessels because of its sensitive detection of hemoglobin in the NIR window. Furthermore, PAI provides multiparametric information and high-resolution vascular structures by using the distinct light absorption characteristics of different biomolecules [56,104].

First, high-resolution vascular imaging is particularly useful for intuitively observing drug-induced vasoconstriction and vasodilation. Kim et al. quantitatively monitored the vasoconstriction induced by corticosteroid treatment in mouse skin using PA microscopy (PAM) [81]. The study showed vasoconstriction in the papillary and reticular dermis when applied topically, compared to the subcutaneously injected model (Figure 2a). Ahn et al. observed vasoconstriction induced by the intraperitoneal injection of glucose caused by acute hyperglycemia in mouse ears [105] (Figure 2b). Huda et al. captured acute vascular reactions in the systemic vessels and placenta of pregnant mice using two types of vasodilators (sildenafil and G protein-coupled receptor G-1) with a PA computed tomography system (PACT) [106] (Figure 2c). These studies directly observed size changes in individual vessels using high-resolution PAI.

Various studies have expanded beyond analyzing individual vessels to multiparametric changes in the vascular system in response to drug administration, along with the analysis of physiology and functional dynamics. Specifically, these studies targeted specific organs, such as the brain, tumors, and other biological tissues, to effectively observe and quantify these changes. First, changes in the cerebrovascular system can be early signs of neurological disorders, strokes, or brain tumors [107,108]; thus, these changes are observed using PAI. Zhu et al. observed both vasoconstriction and vasodilation along with sO_2_ analysis in real-time in the entire cortex of a mouse brain in response to sodium nitroprusside (SNP) through a femoral vein catheter [82]. Using ultrafast wide-field PAM, they obtained high-resolution vascular images and quantitatively analyzed vessel diameter, vessel area, and average sO_2_ in the arteries and veins following SNP administration (Figure 3a,b). Another study observed the cerebrovascular system’s response to SNP [109]. This study used functional PAM to quantitatively analyze the differences in arterial and venous vasodilation as well as the reduction in sO_2_ and blood pressure in both the mouse brain and skin. Shan et al. used real-time PACT to non-invasively monitor the effects of acute prenatal ethanol exposure on the fetal brain vasculature and hemodynamics [110]. This study revealed a decrease in fetal brain vessel diameter, perfusion, and sO_2_ after ethanol exposure. Zhang et al. investigated the reactivity of cerebral microvessels after intravenous epinephrine injection [111]. This study quantitatively analyzed vasoconstriction in brain microvessels, a complementary increase in large vessels, and a decrease in oxy-hemoglobin levels within the entire cerebral vasculature 6 min after administration.

Second, PA has also been instrumental in evaluating the response of tumors to vascular-targeted therapies. By providing real-time information on blood vessel changes, sO_2_ levels, and blood flow within the tumor, PAI aids in assessing drug effects, therapeutic decisions, and prognoses [112,113]. Johnson et al. used a high-resolution all-optical PA scanner to monitor the response to the vascular-disrupting agent OXi4503 [112]. In this study, PAI qualitatively and quantitatively analyzed the changes in tumor vasculature induced by OXi4503 in different carcinoma models in a dose-dependent manner. Yang et al. assessed early tumor responses through changes in total hemoglobin and deoxy-hemoglobin levels by PAM, comparing low- and high-dose groups of intraperitoneally injected bevacizumab, which is used for anti-angiogenic therapy [114]. Fadhel et al. used PAI to monitor the effect of the vascular disrupting agents 5,6-Dimethylxanthenone-4-acetic acid (DMXAA) on tumors [115]. In the study of unmixed oxy-hemoglobin, deoxy-hemoglobin, and methemoglobin and in DMXAA-treated mice, an increase in deoxy-hemoglobin and a decrease in total hemoglobin concentrations were observed, which are consistent with the characteristics of DMXAA that cause vascular disruption. Neuschmelting et al. [83] conducted a study to evaluate the effectiveness of handheld multispectral optoacoustic tomography (MSOT) for detecting vascular-targeted photodynamic therapy (VTP) with the photosensitizer WST11 in a renal cancer mouse model. The study non-invasively tracked changes in sO_2_ for the first hour and 24 and 48 h, after administration. They observed that within the first hour, tumor vasculature destruction was evident in the WST11-VTP group compared to the control group (saline), accompanied by a 60% reduction in tumor sO_2_ levels (Figure 3c). In addition, the sO_2_ levels decreased at 24 and 48 h, demonstrating that MSOT can successfully monitor VTP.

Third, the application of PAI has been extended to other biological tissues in animal study, such as the placenta and kidney. Longitudinal intravital imaging of the mouse placenta following alcohol exposure was observed by analyzing significant changes in sO_2_, vessel diameter, and density by Zhu et al. [84] (Figure 4a). They observed vasodilation in the arteries, and hyperperfusion and elevated sO_2_ levels were confirmed following alcohol consumption using ultrafast functional PAM through an implantable placental window (Figure 4b). Sun et al. simultaneously assessed sO_2_, blood flow, and the metabolic rate of oxygen in peritubular capillaries in acute kidney injury following lipopolysaccharide-induced sepsis [116].

Finally, this approach was further extended to human studies. Bunke et al. monitored the local changes in sO_2_ following adrenaline injection in the human forearm [117]. This study provided real-time information on vasoconstriction and sO_2_ variations, demonstrating the capability of PA to capture dynamic physiological responses in human skin tissues (Figure 4c). Petri et al. observed changes in sO_2_ after applying a hemoglobin spray to chronic leg ulcers [118]. By quantitatively comparing the increase in sO_2_ at 5 and 20 min after spray application, they validated the ability of the hemoglobin spray to enhance oxygenation. Label-free PAI has been successfully employed across preclinical and clinical fields to analyze the vascular and physiological changes induced during drug delivery.

## 4. Photoacoustic Monitoring of Pharmacokinetics and Biodistribution of Exogenous Agents

As aforementioned, PAI can be used to monitor changes in the vascular system that occur during drug delivery. In addition, PAI can be used to perform contrast-enhanced imaging using drugs or chemical materials as contrast agents. Various exogenous agents, such as small-molecule dyes, organic particles, metallic nanoparticles, and polymer-based nanomaterials, have been used in PAI. Indocyanine green (ICG) dye, approved by the Food and Drug Administration (FDA) and widely used in FLI, is the most basic and commonly used agent. As demonstrated in previous studies [80,119], ICG is widely used in feasibility tests to validate the contrast-enhanced performance or pharmacokinetic properties of newly developed PAI systems (Figure 5a). Recently, ICG has been modified to enhance its targeting capabilities and improve various functions because monomeric ICG has low photostability and targeting capabilities [68,120,121].

Singh et al. [122] developed a size-tunable ICG-based platform through J-aggregates (JAs) made of ICG. Compared with monomeric dyes, JAs form head–tail arrangements of monomeric dyes and have greater photothermal stability, a stronger PA signal, and a strong red shift in the absorbance wavelength. Conventional ICG-JAs are formed by self-assembly, often resulting in micron-sized polydisperse aggregates. Therefore, nanoscale ICG-JAs require filtration or encapsulation, which limits their flexibility, including targeting modifications. In this study, a controllable azide-modified ICG-JA (JAAZ) was proposed. The JAAZ is 230–1200 nm in size and can be directly functionalized using NIR absorption and strong PA signals. The developed JAAZ also exhibited a red-shifted absorption peak at 895 nm, similar to the general JA of ICG (Figure 5b). Furthermore, they modified the JAAZ to be functionalized with biomolecules and targeting moieties such as streptavidin and biotinylated RGD (Figure 5c). This functionalized JAAZ showed great performance in PA enhancement compared to the general ICG (ICG vs. RGD-JAAZ, CNR: 1.59 vs. 2.42, circulation half-life (min): 1.19 vs. 19.26) (Figure 5d).

Similarly, small-molecule dyes with high biocompatibility and low toxicity are widely used in PAI and fluorescence imaging. Recently, Yan et al. [123] proposed the development of ultraphotostable small-molecule dyes to promote near-infrared biophotons. These dyes use a ground-state antiaromatic strategy to enhance their performance and achieve ultrafast excited-state dynamics and unmatched photostability. These dyes, with their ultraphotostable properties and broad spectral coverage from 700 to 1600 nm, exhibit high sensitivity and specificity for FLI, making them ideal for detailed cellular and molecular studies. Specifically, this study highlighted the application of these dyes in a traumatic brain injury (TBI) model, showing their PA ability to effectively provide detailed imaging of the ONOO^−^ overproduction area. As a result of unmixing the PA signals, Figure 5e shows an 8-fold increase in the deoxy-hemoglobin signal, a 2.4-fold decrease in the oxy-hemoglobin signal in the TBI area compared with the normal area, and a 6-fold increase in the dye signal. This dual functionality in both the FL and PA modalities underscores the versatility and potential of these small-molecule dyes.

Rathnamalala et al. [124] developed xanthene-based nitric oxide (NO)-responsive nanosensors for PAI in the shortwave infrared (SWIR) window. These dyes were characterized by their ability to absorb light from 900 to 1400 nm. The developed SCR-1 dye (SWIR xanthene dye based on a dibenzazepine donor conjugated to thiophene) was used to develop a ratiometric nanoparticle for NO (rNP-NO), which enabled the successful visualization of pathological levels of nitric oxide in a drug-induced liver injury model by deep tissue SWIR PAI. The SWIR PA signals from the reference dye IR-1061 were detected in both the control and treatment groups, but the signals from SCR-NO (NO-responsive molecule) were obtained only in the control group.

Metallic nanoparticles provide distinct benefits, whereas small-molecule dyes offer great advantages in both FL and PA imaging owing to their broad spectral coverage and photostability. Metallic nanoparticles, known for their strong optical absorption and scattering properties, enhance image contrast more effectively in deep tissue imaging and exhibit greater stability under physiological conditions [125]. These unique properties make metallic nanoparticles a powerful alternative to PAI, particularly for applications requiring high sensitivity and long-term stability.

Gold nanoparticles are representative PAI contrast agents [126]. Sun et al. [127] recently developed gas-generating laser-activatable nanorods for contrast enhancement (GLANCE), which significantly improved the performance of US and PA imaging. The GLANCE system used the photocatalytic properties of gold nanorods to generate nitrogen gas bubbles upon NIR laser irradiation, which enhanced the acoustic impedance contrast in the targeted tissues. Figure 6a shows the difference in the mapping of the US and PA images before and after laser irradiation. In both the US and PA images, only the irradiated region was represented.

In addition to gold nanoparticles, iron-based nanoparticles are another promising type of metal nanoparticle used in PAI. Zhao et al. [85] developed NIR phototheranostic iron pyrite (FeS_2_) nanocrystals for enhanced imaging and therapy in triple-negative breast cancer (TNBC). These iron pyrite nanocrystals, when exposed to NIR light, exhibit high photothermal conversion efficiency (63.1%), making them highly effective for PAI and photothermal therapy (PTT). The study demonstrated that upon NIR irradiation, FeS_2_ nanocrystals significantly enhanced PA signals and induced dual cell death pathways, apoptosis and ferroptosis, by accelerating Fenton reactions and increasing ROS production. The FeS_2_ nanocrystals were modified with PEG to improve their biocompatibility and stability, resulting in a strong PA signal and efficient tumor targeting (Figure 6b). In TNBC models, these nanocrystals facilitated tumor-targeting imaging and effectively suppressed tumor growth and metastasis.

Furthermore, polymer-based nanomaterials showed great potential for use in PAI [128]. These nanomaterials have several unique advantages, including high biocompatibility, tunable physical and chemical properties, and the ability to carry multiple imaging and therapeutic agents. Lorenz et al. [129] introduced a novel polymeric nanoparticle specifically designed for the diagnosis of ovarian cancer. This polymeric nanoparticle loaded with two naphthalocyanine dyes provided a distinct spectral signature with two well-separated PA signal peaks at 770 and 860 nm. These peaks enabled precise spectral unmixing, distinguishing nanomaterial signals from endogenous chromophores in cancer tissues. In vivo experiments in mice with subcutaneous and intraperitoneal ovarian cancer xenografts confirmed high-contrast tumor visualization and accurate diagnosis. Furthermore, this nanoagent demonstrated effective PTT capabilities, selectively eradicating ovarian cancer cells by elevating the intratumoral temperature to approximately 49 °C upon NIR light exposure.

Song et al. [86] developed a semiconducting homopolymer nanoplatform designed for NIR PAI and photo-triggered treatment of thrombosis. This nanoplatform integrated a fibrin-specific homopolymer with strong NIR-II absorption and a thermosensitive NO donor. The fibrin-specific ligand ensured selective targeting of thrombi, whereas the NO donor enabled localized therapy. Upon exposure to NIR-II light, these nanoprobes exhibited bright PA signals and efficiently accumulated in the thrombi, allowing sensitive and selective thrombus delineation (Figure 6c). In addition, the nanoplatform demonstrated rapid and effective thrombus removal by combining photothermal effects with on-demand NO release, restoring blood flow in both the carotid and lower-extremity arterial thrombosis models.

## 5. Discussion

PAI has emerged as a powerful biomedical imaging tool that combines the advantages of optical and ultrasound imaging. PAI uses various molecules as contrast agents, including endogenous and exogenous chromophores. These chromophores exhibit specific light absorption properties, allowing PAI to capture and monitor vascular structures, hemodynamics, sO_2_, and the distribution of materials with high spatial resolution. This multiparametric information is obtained in a non-invasive manner, making PAI invaluable for both preclinical and clinical applications.

Based on its label-free imaging capability, PAI facilitates the observation of morphology, hemodynamics, and physiology following drug delivery without directly detecting contrast agents. By leveraging the distinct light absorption characteristics of biomolecules, PAI can provide high-resolution vascular structures and multiparametric information. PAI has been used for the intuitive observation of vasoconstriction and vasodilation induced by various drugs based on high-resolution vessel images. Moreover, various studies have shown the capability of evaluating the dynamic response to drug delivery in the brain, tumors, placenta, kidneys, and human tissues.

In addition, PAI plays a significant role in directly tracking the biodistribution and pharmacokinetics of administered contrast agents. The use of small-molecule dyes, such as ICG, and the development of small-molecule-dye-based modified agents have resulted in significant improvements in PAI performance and targeting capabilities. Other contrast agents, including metallic nanoparticles and polymer-based nanomaterials, have enhanced their capability to provide strong optical absorption, scattering properties, and stability under physiological conditions.

Collectively, these studies demonstrate the significant role of PAI in understanding reactivity after drug delivery. By facilitating real-time assessment of vascular and functional changes, as well as the biodistribution of contrast agents, PAI can support both early diagnosis and personalized treatment strategies. Table 3 shows the summary of studies using PAI to monitor drug delivery responses. 

Further advancements in PAI technology are essential for future applications in comprehensive monitoring following drug delivery. First, an advanced multimodal system with other conventional modalities, such as US, MRI, PET, and CT, can provide complementary information, thereby enhancing the monitoring capabilities of PAI. This multimodal approach can merge the strengths of each modality, offering a more detailed and comprehensive assessment of biological processes [130,131,132]. Moreover, deep learning-based methods can overcome the challenges of PAI, such as limited bandwidth, limited view, and assumptions in processing [80,87,133,134,135,136]. In conclusion, further studies to overcome these challenges will enable PAI to more effectively track detailed drug delivery responses, making it a valuable tool in the preclinical and clinical fields.

## Figures and Tables

**Figure 1 pharmaceutics-16-01240-f001:**
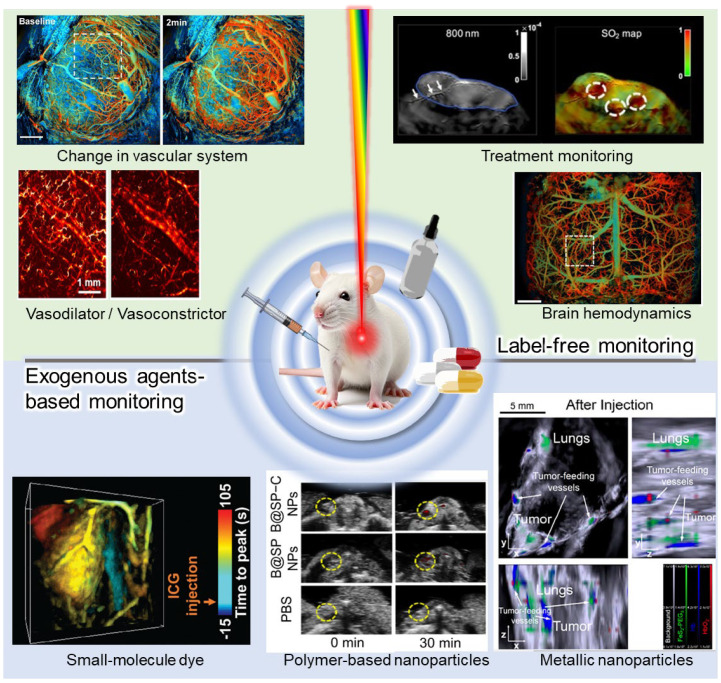
Summary of photoacoustic (PA) monitoring after drug delivery, representing label-free PA monitoring and exogenous agent-based monitoring. In this review, we first discuss studies that observed changes in the vascular system and individual vessel responses to drug delivery (top left). Then, we highlight label-free PA treatment monitoring and brain hemodynamics, including oxygen saturation mapping and associated hemodynamic changes post-drug administration (top right). Finally, we review exogenous agent-based PA monitoring, categorizing the agents used as follows: small-molecule dye, polymer-based nanoparticles, and metallic nanoparticles (Bottom). SO_2_, oxygen saturation. The images are adapted with permission from [80,81,82,83,84,85,86]. Copyright 2023 American Chemical Society.

**Figure 2 pharmaceutics-16-01240-f002:**
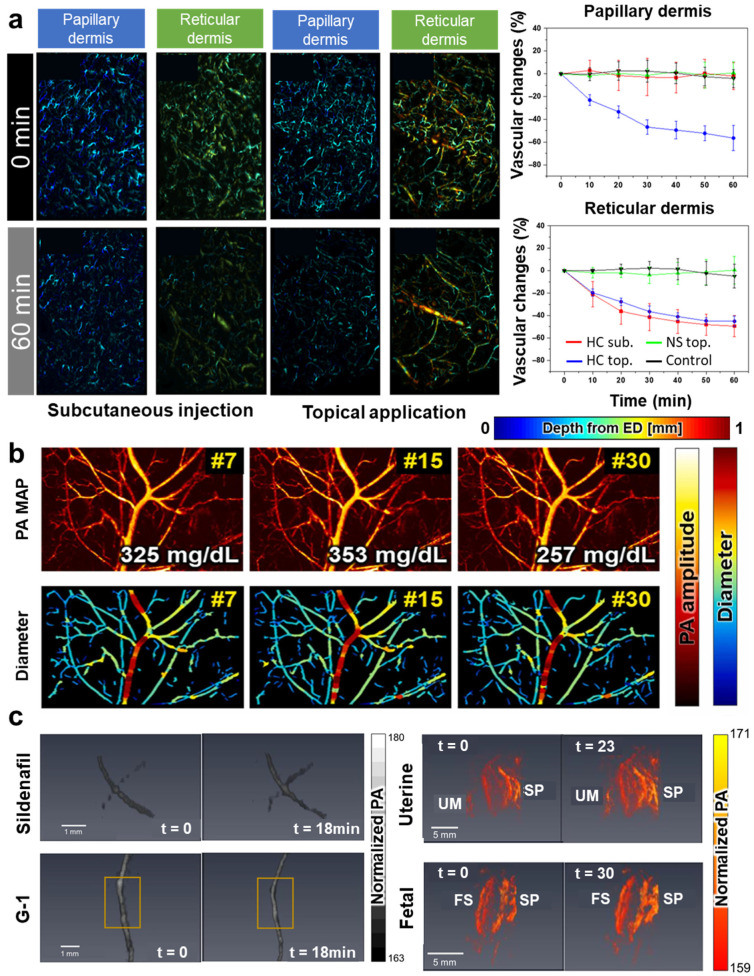
(**a**) PA images of the vasoconstriction after the subcutaneous injection and topical application of hydrocortisone, and quantified vascular changes in the papillary dermis and reticular dermis. (**b**) PA maximum amplitude projection (MAP) images and diameter-mapped images after the injection of different glucose concentrations at 20-min (#7), 60-min (#15), and 135-min (#30) post-injection. (**c**) Volume images of the iliac artery after injection of sildenafil and internal thoracic artery after injection of G-1 (**left**), and images of uterine artery and fetal vasculature after injection of vasodilators (**right**). HC, hydrocortisone; NS, nonsteroidal; sub., subcutaneous injection; top., topical application; ED, epidermis; PA, photoacoustic; MAP, maximum amplitude projection; UM, umbilical cord; SP, spiral artery; and FS, fetal side. The images are adapted with permission from [81,105,106].

**Figure 3 pharmaceutics-16-01240-f003:**
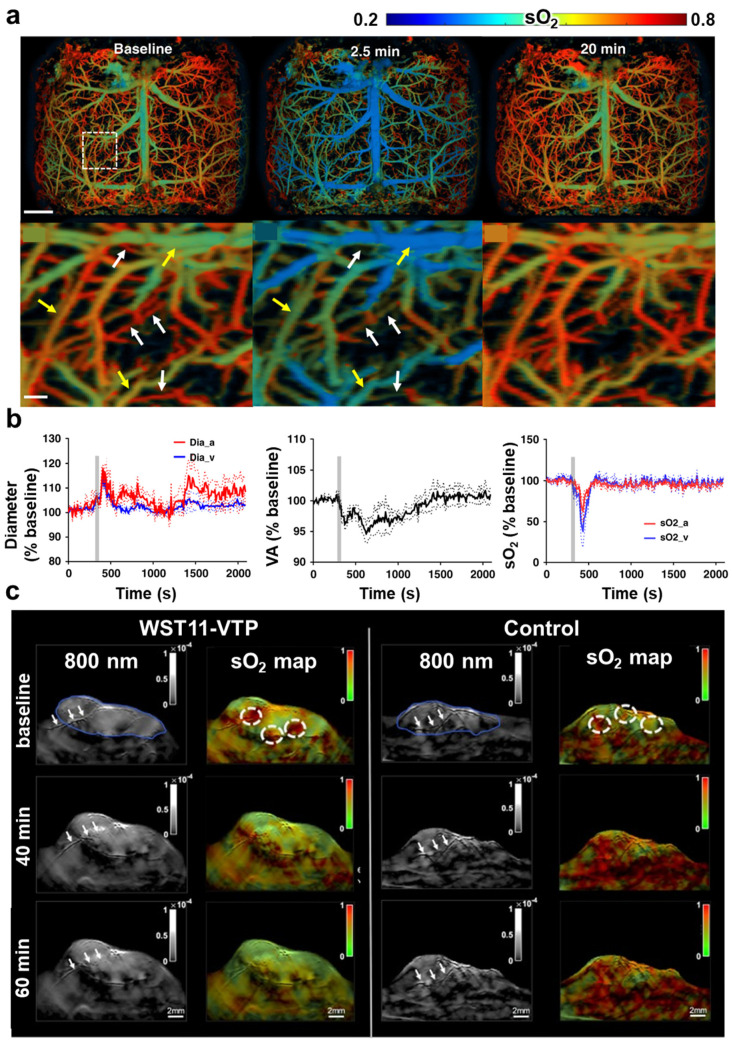
(**a**) Oxygen saturation images of a mouse brain in response to sodium nitroprusside. The images visualize individual vessels showing vasoconstriction (white arrows) or vasodilation (yellow arrows). The white box indicates a magnified region, which is enlarged for the same area at all time points. Scale bar, 1mm (top) and 100 μm (bottom). (**b**) Quantitative vessel diameter, area, and oxygen saturation changes after SNP injection (gray bar). Data are shown by mean ± standard error of the mean. (**c**) PA images of the first 60 min after WST11 administration to induce VTP. White arrows indicate that the vessel disrupts after VTP, and white dashed circles show regions in which sO_2_ levels decreased. sO_2_, oxygen saturation; VA, vessel area; SNP, sodium nitroprusside; VTP, vascular-targeted photodynamic therapy. The images are adapted with permission from [82,83].

**Figure 4 pharmaceutics-16-01240-f004:**
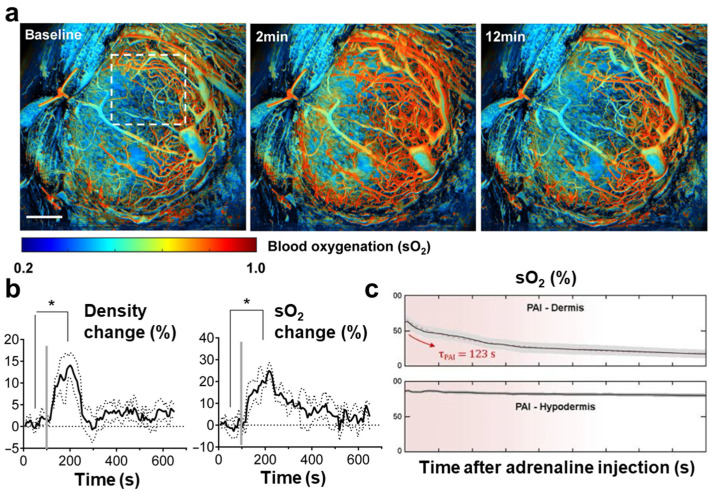
(**a**) Ultrafast functional PAM images of a placenta at baseline, 2 min, and 12 min after alcohol consumption. Scale bar, 1 mm. (**b**) Changes in vessel density and oxygen saturation in the placenta after alcohol consumption (N = 3). The gray bars indicate the start of alcohol administrations (100 s). Data are shown by mean ± standard error of the mean. (**c**) Changes in the oxygen saturation level of the dermis and hypodermis after adrenaline injection in the human forearm. * *p* < 0.05; sO_2_, oxygen saturation. The images are adapted with permission from [84,117].

**Figure 5 pharmaceutics-16-01240-f005:**
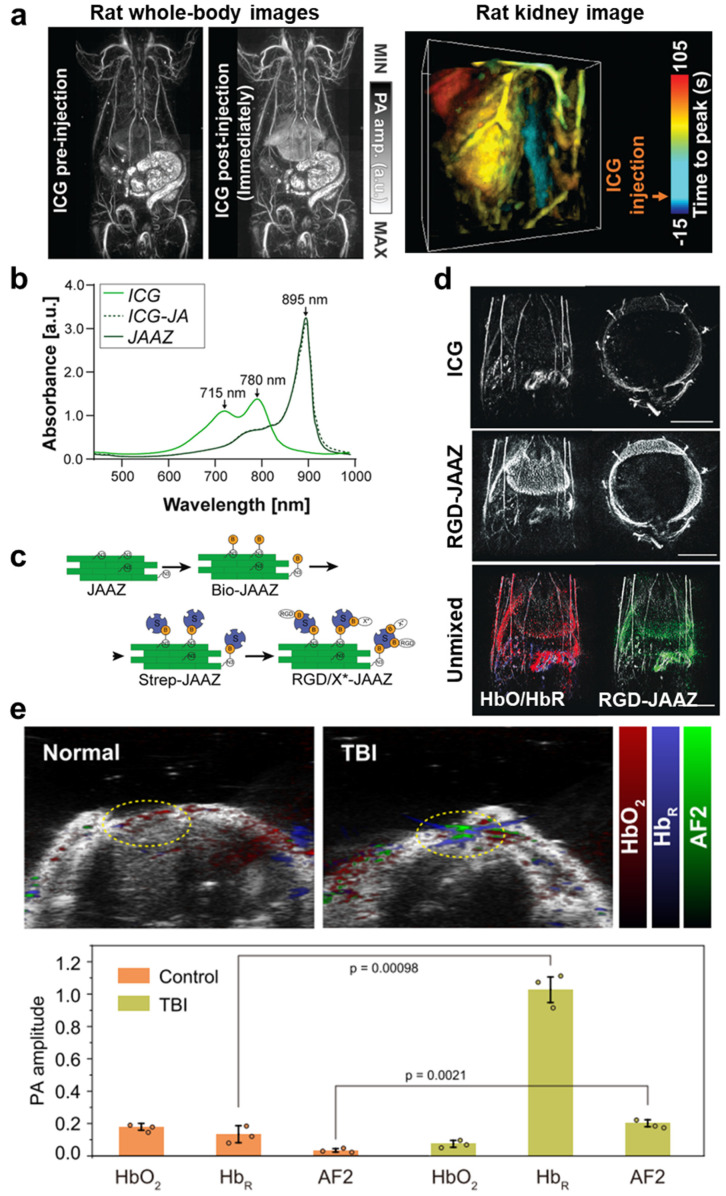
(**a**) PA MAP images of a whole-body rat before and after ICG injection and 3D time-to-peak images of ICG perfusion in the rat kidney. (**b**) Absorbance of ICG, ICG-JA, and JAAZ particles at 50 µM. (**c**) Schematic of the functionalization of JAAZ particles coating with biomolecules. (**d**) PA MAP images of mice after ICG and RGD-JAAZ injection and PA unmixed images of HbO, HbR, and RGD-JAAZ. Red colormap shows HbO, blue shows HbR, green shows RGD-JAAZ, and gray shows PA signal. (**e**) US and PA unmixed images of HbO, HbR, and AF2 and comparison of PA signals between the control and TBI models (top). The bar graphs shows statistical (N = 3) PA amplitudes of HbO, HbR, and AF2 after spectral unmixing. PA, photoacoustic; MAP, maximum amplitude projection; ICG, indocyanine green; JAs, J-aggregates; JAAZs, azide-modified ICG J-aggregates; US, ultrasound; TBI, traumatic brain injury; HbO, oxy-hemoglobin; HbR, deoxy-hemoglobin; and AF, aminofluorene. The images are adapted with permission from [80,122,123].

**Figure 6 pharmaceutics-16-01240-f006:**
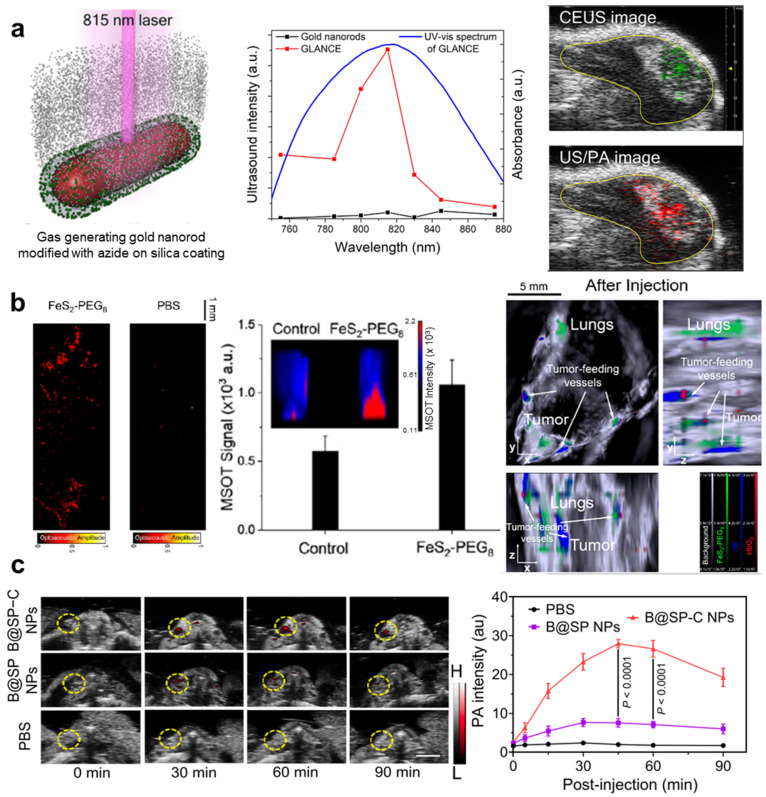
(**a**) Schematic of gas-generating gold nanorod modified with azide on silica coating (**left**). GLANCE’s US signal peaked at 815 nm, matching its SPR peak, while the gold nanorods showed no signal change with varying laser wavelengths (**center**). US and PA images of the tumor corresponding to the laser irradiation (**right**). Green color mapping shows the difference in US intensity between right after and before laser irradiation, and red colormap represents the difference in PA intensity. (**b**) Comparison of PA images of MDA-MB-231 cells between FeS_2_-PEG8 and PBS 24 h uptake (**left**). PA images of xenograft mice bearing MDA-MB-231 breast tumors 1 h after FeS_2_-PEG8 injection (**right**). (**c**) PA and US images of the thrombotic artery after PBS, B@SP NPs, or B@SP-C NPs. Yellow dashed circles represents the thrombotic artery areas. The corresponding PA signals of the thrombotic artery (n = 5). US, ultrasound; SPR, surface plasmon resonance; PA, photoacoustic; NP, nanoparticle. The images are adapted with permission from [85,86,127]. Copyright 2023 and 2024 American Chemical Society.

**Table 1 pharmaceutics-16-01240-t001:** List of abbreviations and acronyms used in this paper.

Abbreviation	Explanation
HbO	Oxy-hemoglobin
HbR	Deoxy-hemoglobin
HbT	Total hemoglobin
sO_2_	Oxygen saturation
MAP	Maximum amplitude projection
SNP	Sodium nitroprusside
DMXAA	5,6-Dimethylxanthenone-4-acetic acid
VTP	Vascular-targeted photodynamic therapy
ICG	Indocyanine green
JA	J-aggregates
SWIR	Shortwave infrared
NO	Nitric oxide
ROS	Reactive oxygen species
NIR	Near infrared

**Table 2 pharmaceutics-16-01240-t002:** Examples of the performance of various types of PAI system.

System	Detector Specification (Center Frequency)	Lateral Resolution	Imaging Depth	Approximate Imaging Area (Target)	Ref.
OR-PAM	Single element (50 MHz)	0.4–0.7 µm	0.76 mm	15 × 10 mm^2^ (Mouse ear)	[98]
	Single element (n/a)	1.2 µm	1 mm	6 × 10 mm^2^ (Mouse brain)	[99]
AR-PAM	Single element (50 MHz)	52.5 µm	3–6 mm	30 × 35 mm^2^ (Mouse ear)	[100]
	Single element (50/75 MHz)	84/54 µm	2.7/1.8 mm	9 × 10 mm^2^ (Mouse ear)	[101]
PACT	1024 elements Hemispherical (2 MHz)	380 µm	10 mm	65 × 85 mm^2^ (Mouse whole body)	[80]
	1024 elements Arc (2.25 MHz)	370–390 µm	40 mm	75 × 85 mm^2^ (Human breast)	[103]

**Table 3 pharmaceutics-16-01240-t003:** Summary of studies that employ photoacoustic imaging monitoring after drug delivery. LRes, lateral resolution; ARes, axial resolution; ID, imaging depth; SNP, sodium nitroprusside; UFF, ultrafast functional; PAM, photoacoustic microscopy; OR, optical-resolution; PACT, photoacoustic computed tomography; AF, aminofluorene; AR, acoustic-resolution; DMXAA, 5,6-Dimethylxanthenone-4-acetic acid; FeS_2_, iron pyrite; NO, nitric oxide; JAAZ, modified ICG J-aggregate; ICG, Indocyanine green, and n/a, not applicable.

Target Tissue	Image Modality	Detector Spec. (Center F)	Imaging Performance	Detection	Drug or Contrast Agents	Ref.
Skin	OR-PAM	Single (50 MHz)	LRes: 5 µm ARes: 30 µm ID: 1 mm	Vasoconstriction	Corticosteroid	[81]
Ear	PAM	Single (50 MHz)	LRes: 5 µm IDes: 1 mm	Vasoconstriction	Glucose	[105]
Placenta	PACT	96 el. Arc (6 MHz)	LRes: 390 µm ARes: 370 µm	Vasodilation	Sildenafil and G protein-coupled receptor G-1	[106]
	UFF-PAM	Single (40 MHz)	LRes: 10 µm	Vasodilation Vascular structure Oxygenation	Alcohol	[84]
Brain	UFF-PAM	Single (40 MHz)	LRes: 10 µm IDes: 1.5 mm	Vasoconstriction Vasodilation Oxygenation	SNP	[82]
	OR-PAM	Single (30 MHz)	LRes: 3 µm ARes: 25 µm	Vasodilation Oxygenation	SNP	[109]
	PACT	256 el. Cylinder (4 MHz)	n/a	Vascular structure Oxygenation	Alcohol	[110]
	PAM	n/a	ID: several millimeters	Vasoconstriction Oxygenation	Epinephrine	[111]
	PACT	256 el. Linear (n/a)	n/a	Agent’s biodistribution Oxygenation	AF-based dye	[123]
Tumor	All-optical PA	Fabry–Perot sensor	LRes: 50–150 µm ARes: 50–150 µm ID: 10 mm	Vascular structure	OXi4503	[112]
	AR-PAM	Single (25 MHz)	LRes: 130 µm ARes: 60 µm	Vascular structure Oxygenation	Bevacizumab	[114]
	PACT	Linear (15 MHz)	n/a	Oxygenation	DMXAA	[115]
	PACT	256 el. Arc (4 MHz)	Res: 200 µm	Vascular structure Oxygenation	WST11	[83]
	PACT	Array (40 MHz)	n/a	Agent’s biodistribution	Gas-generating laser-activatable nanorods	[127]
	PACT	256 el. Array (5 MHz)	n/a	Agent’s biodistribution	FeS₂ nanocrystals	[85]
	PACT	Array (40 MHz)	ARes: 40 µm	Agent’s biodistribution	Polymeric nanoparticle	[129]
Human forearm skin	PACT	Array (30 MHz)	LRes: 50 µm ARes: 110 µm IDes: 20 µm	Vasoconstriction Oxygenation	Adrenaline	[117]
Ceritubular capillary	PAM	Single (35 MHz)	IDes: 200 µm	Vascular structure Oxygenation	Lipopolysaccharide	[116]
Human leg	PACT	Linear array (21 MHz)	n/a	Oxygenation	Hemoglobin spray	[118]
Whole body	PACT	1024 el. Hemispherical (2 MHz)	Isotropic R: 380 µm	Agent’s biodistribution	ICG	[80]
Liver and spleen	PACT	Single (5, 35 MHz)	ID: 5 mm	Agent’s biodistribution	JAAZs	[122]
	PACT	128 el. Array (5 MHz)	n/a	Agent’s biodistribution	Xanthene-based NO-responsive nanosensors	[124]
Carotid artery	PACT	Array (n/a)	n/a	Agent’s biodistribution	Semiconducting homopolymer nanoplatform	[128]

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
