# Peer review of "Review on Photoacoustic Monitoring after Drug Delivery: From Label-Free Biomarkers to Pharmacokinetics Agents"

_pharmaceutics, 2024, doi:10.3390/pharmaceutics16101240_

Round 1
Reviewer 1 Report
Comments and Suggestions for Authors
This paper explores the application of photoacoustic imaging with a label-free method to track how drugs are delivered in the vascular system and oxygen saturation levels in different biological tissues. Furthermore, photoacoustic imaging with exogenous contrast agents is proven to provide enhanced imaging of affected areas. The article is well structured and well written. This work can be published with the resolution of several minor questions below.
(1) Fig. 1, more detailed information about the content of each picture can be added.
(2) Page 5, it would be better if authors reinterpret HbO and HbR or simply use their full names.
(3) Page 9, it seems that these two paragraphs are both about the application of PAI in other biological tissues. I am confused why these two paragraphs are not classified as the same but in parallel.
(4) Table 1, this article categorizes photoacoustic after drug delivery into label-free biomarkers and pharmacokinetics agents, why didn't authors add such a classification in the table.
Comments on the Quality of English Languagewell written and comprehensible
Reviewer 2 Report
Comments and Suggestions for Authors
Review on Photoacoustic Monitoring after Drug Delivery: From Label-Free Biomarkers to Pharmacokinetics Agents by Kim et al.
The paper presents a literature survey on photoacoustics-based imaging techniques applied to some pharmaceutical studies. The paper is understandable, but language needs some editing. Although the structure of the item is quite standard - it consists of consecutive introduction (1), theoretical background (2), data analysis (3-4) and discussion/conclusion (5) sections, the text itself is quite chaotic, it lacks of a common background and a some sort of continuity between the sections. To be well understood – section 2 provides some general concepts on the PA method, section 3 gives some literature examples on sO2 monitoring for bio-imaging purposes (but the content is barely related to photoacoustics-relevant topics, as suggested by the paper title; it is just a list of studies related to haemoglobin monitoring, without pointing at photothermally relevant properties allowing for the measurements), then section 4 provides a survey on various (probably) photothermally-beneficial setups for PA imaging, however, a direct relation to photoacoustics or photothermal properties of systems still remains blurry. Considering the sections content, the paper is unlike to be a scientifically-friendly for a broader audience, especially for early stage scientists and others without background in photoacoustics and imaging techniques in the same time. Moreover, even for the narrow audience mentioned, there is an insufficient amount concise information on photothermal properties of the systems reviewed. Eventually, I think that the paper can be considered for publication only after heavy and thorough rearrangements and refinements.
What I suggest to start with, is to refine the section 2. Beside some information on PA (as it is now), it would be convenient to add information about experimental setups and configurations (common light sources, detectors, signal creation, detections components (f.e acoustic lenses), preamplifiers etc) for all of the PAI modalities mentioned in the text, and, what is even more important for the paper, conditions/characterization of a good photothermally-active samples, which will underline the capabilities/useability of the methods and the idea behind photothermal characterization of samples. As is for now, the authors only mention that the “initial pressure” (pressure perturbation) depends on the fluence (source), absorption coefficient and nonradiative deactivation efficiency (PA-active specimens - pigments) and Gruneisen parameter (pigments/environment). However, the P0 is also proportional to the PA response (which is crucial to understand the paper), so the PA magnitude can be generally magnified or tailored by the experimental conditions (the light source intensity, but this can be also harmful for the samples), the pigment concentration (can be toxic for organisms) or chemical modification of the pigments (increasing of the nonradiative deactivation efficiency). Moreover, the pigments have to display spectral characteristics to be differentiable from the environment. These properties should be underlined and their actual values analysed throughout the paper for subsequent systems described (f.e. – show PA spectra of HbO and HbR – pure and for real (blood) sample, identify the absorption maxima, nonradiative deactivation constants, the same for other exemplary systems from section 4, whenever possible). Also, some sort of a table, summarizing the state-of-the-art PA systems (techniques) and its PA-relevant properties and useability for particular systems (may give a clue how to develop new PA-based imaging systems), would be beneficial for the interested audience.
Other issues:
The introduction is far too general (too much: “restricted”, “low” etc). The authors should underline (once again, maybe a table?) some crucial characteristics of the PA imaging modalities mentioned in the text (i.e. actual and up to date: resolution ranges, scanning depths, applicability range, cautions etc), especially in relation to other imaging techniques. Are there any areas where PAI is unique in some sort?
Fig.1 (but also other figures) – a lot of details (arrows, frames) without explanation/description. The authors must consider, that the readers may not have an access to the cited papers, and all of the figures (but also research behind the graphics) adapted from other papers should be precisely described in the text.
Eq.2 – The PA signal is not equal to the product of the absorption coefficient and concentration; it is proportional to the product. What I suggest is to relate eq.1 to the PA signal magnitude (via proportionality sign), and then give PA signal of “multispectral” sample by a sum of single components contribution given by eq.1.
The third equation is also numbered as eq.2.
In general, the paper suffers from enormous amounts of abbreviations. Whenever possible, the authors should use full names of the techniques (maybe except PAI), or create a kind of glossary/list of abbreviations close to the introduction section.
Comments on the Quality of English LanguageThe text is generally understanable, minor revisions required.
Round 2
Reviewer 2 Report
Comments and Suggestions for Authors
In general I'm quite satisfied how the paper progressed. I still believe it would be beneficial to extend the paper in the ways as described in the previous review report - please bare in mind this is a review type of paper, it is hard to be "outside the scope" when describing such a broad topic, while the informations demanded are crucial to understand the status of methodologies described, again, primarly for the sake of interested audience. I think, however, that I turned the authors attention on the issues, and the final decission in the Editor hands (I recommend further, minor revisions).
Comments on the Quality of English LanguageThe paper is understandable.
